# COVID-19 vaccine rumors and conspiracy theories: The need for cognitive inoculation against misinformation to improve vaccine adherence

Md Saiful Islam [1,2¤]*, Abu-Hena Mostofa Kamal [3,4], Alamgir Kabir[2,5], Dorothy L. Southern[6], Sazzad Hossain Khan[1], S. M. Murshid Hasan[7], Tonmoy Sarkar[1], Shayla Sharmin[8], Shiuli Das[1], Tuhin Roy[9], Md Golam Dostogir Harun[1], Abrar Ahmad Chughtai[2], Nusrat Homaira[10], Holly Seale[2]

1 Infectious Diseases Division, Program for Emerging Infections, International Centre for Diarrheal Diseases Research, Dhaka, Bangladesh, 2 School of Population Health, University of New South Wales, Sydney, Australia, 3 Khulna University of Engineering and Technology, Khulna, Bangladesh, 4 Department of Sociology, University of Saskatchewan, Saskatoon, Canada, 5 Centre for Primary Health Care and Equity, University of New South Wales, Sydney, Australia, 6 Independent Scientific Writing Consultant, Mandalay, Myanmar, 7 Department of Society and Health, Mahidol University, Salaya, Thailand, 8 Independent Researcher, Sydney, Australia, 9 Sociology Discipline, Khulna University, Khulna, Bangladesh, 10 School of Women's and Children's Health, University of New South Wales, Sydney, Australia

¤ Current address: School of Population Health, Faculty of Medicine, University of New South Wales, Kensington, Australia
* saiful@icddrb.org

**Data Availability Statement:** All relevant data are within the paper and its Supporting information files.

## Abstract

## Introduction

Rumors and conspiracy theories, can contribute to vaccine hesitancy. Monitoring online data related to COVID-19 vaccine candidates can track vaccine misinformation in real-time and assist in negating its impact. This study aimed to examine COVID-19 vaccine rumors and conspiracy theories circulating on online platforms, understand their context, and then review interventions to manage this misinformation and increase vaccine acceptance.

## Method

In June 2020, a multi-disciplinary team was formed to review and collect online rumors and conspiracy theories between 31 December 2019–30 November 2020. Sources included Google, Google Fact Check, Facebook, YouTube, Twitter, fact-checking agency websites, and television and newspaper websites. Quantitative data were extracted, entered in an Excel spreadsheet, and analyzed descriptively using the statistical package R version 4.0.3. We conducted a content analysis of the qualitative information from news articles, online reports and blogs and compared with findings from quantitative data. Based on the fact-checking agency ratings, information was categorized as true, false, misleading, or exaggerated.

**Funding:** This work did not receive any funding from a donor.

**Competing interests:** Associate Professor Holly Seale has previously received funding from drug companies for investigator-driven research and consulting fees to present at conferences/workshops and develop resources (Seqirus, GSK and Sanofi Pasteur). She has also participated in an advisory board meeting for Sanofi Pasteur. This work has been a voluntary contribution from all authors. The commercial affiliation of the authors did not play a role in our study, and this does not alter our adherence to PLOS ONE policies of sharing data and materials.

## Results

We identified 637 COVID-19 vaccine-related items: 91% were rumors and 9% were conspiracy theories from 52 countries. Of the 578 rumors, 36% were related to vaccine development, availability, and access, 20% related to morbidity and mortality, 8% to safety, efficacy, and acceptance, and the rest were other categories. Of the 637 items, 5% (30/) were true, 83% (528/637) were false, 10% (66/637) were misleading, and 2% (13/637) were exaggerated.

## Conclusions

Rumors and conspiracy theories may lead to mistrust contributing to vaccine hesitancy. Tracking COVID-19 vaccine misinformation in real-time and engaging with social media to disseminate correct information could help safeguard the public against misinformation.

## Introduction

Rumors and conspiracy theories, have been identified as precipitators for vaccine hesitancy [1]. People may decline immunizations due to false claims that vaccines contain infertility agents or can spread an infectious pathogen such as human immunodeficiency virus (HIV) [2, 3]. Historically, negative claims about vaccine effectiveness have affected vaccine uptake. A boycott of the polio vaccine due to rumors that the vaccine caused infertility led to increased polio cases in Nigeria, Pakistan, and Afghanistan [2, 4]. Rumors often challenge the health policies and interventions of government and non-government officials and international health agencies such as the World Health Organization (WHO) [5]. Whether a person believes the misinformation or not is dependent on the individual's level of health literacy and their risk perceptions. However, continuous exposure to social media and online anti-vaccine movement may influence people to share and communicate vaccine misinformation and conspiracy theories [6–8].

Social media platforms have become a common source for health information. During a pandemic, people may use social media to improve their knowledge about the disease, transmission, and prevention mechanisms [9, 10]. Health information circulating on online platforms are often amplified by rumors and conspiracy theories that are not always based on scientific evidence [11]. Health information-seeking behavior on online platforms puts users at risk of being exposed to misinformation that could potentially threaten public health [12]. A 2020 study conducted by Stecula and colleagues found that people who were exposed to vaccine-related information on social media were more likely to be misinformed and become vaccine-hesitant [13]. A separate study of 2000 adults in the United Kingdom (UK) conducted by the Royal Society for Public Health, found that two-fifths of the participants encountered negative messages about vaccination on social media platforms [14]. People often share concerns, mistrust and rumors about vaccines on social media before they are detected through a traditional surveillance system, such as event-based surveillance [15]. Therefore, monitoring this media data has been identified as one of the best methods for tracking misinformation in real-time and as a possible way to dispel misinformation and optimize vaccine acceptance.

Given that in early 2021 there are at least four vaccine candidates that have been approved for emergency use against COVID-19 with more options planned for approval, it is critical that efforts are made to start tracking the misinformation being spread [16]. Lack of

confidence in COVID-19 vaccine candidates may lead some people to delay or refuse the new vaccine, potentially disrupting national and international control efforts [17].

Culturally compelling and context-appropriate information, supported by credible sources, are required to prevent vaccine misinformation [18]. This study aimed to examine COVID-19 vaccine rumors and conspiracy theories circulating on online platforms, understand their context, and then review interventions to manage this misinformation and increase vaccine acceptance.

## Materials and methods

### Study design

This was a mixed-method study where we utilized both quantitative and qualitative approaches. In June 2020, a team of social scientists trained in infodemic management and epidemiologists was formed to review and collect COVID-19 vaccine(s) information circulating globally on the online platforms. The team collected the data between 31 December 2019, the day the COVID-19 pandemic was first notified in China, to 30 November 2020.

### Study definitions

We defined rumors as any unverified claims related to COVID-19 vaccine(s) or the process of immunization/vaccination circulating on the online platforms that could be classified as true, false, misleading or exaggerated after verification by the fact-checking agencies [19]. As adapted from Wardle et al. (2021), misinformation was defined as inaccurate or false information shared by someone unwittingly and without any intention to cause harm [20]. We defined conspiracy theories as any claims by an individual or group of people to reach malicious goals [21].

### Data collection

A wide range of sources including Google, Google Fact Check, Facebook, YouTube, Twitter, websites of fact-check agencies, websites of television and newspaper were reviewed. We searched reports in English using the following search terms "2019-nCoV vaccine" or "COVID-19 vaccine" or "2019 novel coronavirus vaccine" and "rumor" or "misinformation" "false information" "fake news" or "conspiracy theories". For each combination of terms searched on Google, we limited our screening to the first 10 pages for evaluation. The fact-check agencies reported vaccine rumors either in English or in local languages and some also published the weblinks of vaccine rumors and conspiracy theories along with screen shots of social media posts on the websites. We visited the websites, reviewed the contents, and documented all the information they reported related to vaccine rumors into our database.

### Data extraction and consistency

Once the vaccine-related items were identified, the team made a list of weblinks. The links were then distributed among nine researchers who had prior experience of extracting data from web-based sources. Each team member visited the allocated weblinks that took the team member to the original content, reviewed the full content of each claim multiple times, and extracted the data into a spreadsheet using a checklist used in a previous infodemiology study. We collected data on the title, country, date, language, social media platform(s) type of information and their factual rating. To measure the magnitude of the spread of each of the news articles, social media narratives and online reports and/or blogs, we collected the number of available likes, shares, emojis and retweets. Once the data were extracted, the team swapped

their weblinks with another member to check for data entry error and consistency. When discrepancies in data entry were found, the lead author met with the data entry team, reviewed the report, and resolved any discrepancies. Once all the data were entered in the spreadsheet, the data were then sorted to check for duplication. Duplicate weblinks, and contents that could not be translated into English, were excluded from the analysis. Finally, we used the Google Trends tool to retrieve data on the internet users' searches related to COVID-19 vaccine(s). We retrieved the data for the study period using the search queries, "COVID-19 vaccine" in English.

## Data analysis

We descriptively analyzed the quantitative data using the open-source statistical package R version 4.0.3 (R Project for Statistical Computing, Vienna, Austria). Stacked bar charts were used to depict the distribution of different news articles, social media narratives, online reports and or blogs by date and country. We also constructed a line chart to show online users search activity related to COVID-19 vaccine on Google. We constructed a global map to describe the geographical distribution of the total count of the rumors and conspiracy theories reported. We also constructed a Venn diagram to show the number of rumors or conspiracy theories shared among different social media platforms. For news articles, online reports, blogs, and fact-check agency reports we conducted content analysis to compare the data collected. The data were then categorized under predefined themes: rumor or conspiracy theory and emergent sub-themes identified by the typology of claims circulating on the online platform, including COVID-19 vaccine development, availability and access, vaccine related morbidity and mortality, political and economic motives, safety, efficacy and acceptance, COVID-19 susceptibility due to exposure to other vaccines, vaccine reagents, mandatory vaccines and ethics, vaccine alternatives and necessity, conspiracy theories and miscellaneous (Table 1). The lead author shared the analysis among all co-authors for review and consensus. Using the fact-check agency ratings, the team also categorized the rumors and conspiracy theories as 'true ', 'false', 'misleading' and 'exaggerated' [22].

**Table 1. Operational definitions of study sub-themes, 31 December 2019–30 November 2020.**

| Sub-themes | Operational definitions |
|---|---|
| Vaccine development, availability, and access | Reports of rumors and conspiracy theories related to the policy, progress, and challenges of COVID-19 vaccine trial, phases of trials, trial participants and development were included under this category. This also includes past and ongoing availability of COVID-19 vaccines, it's delivery and public access to them [34]. |
| Safety, efficacy, and acceptance | Claims related to the safety and efficacy of COVID-19 vaccine and its acceptance |
| Political and economic motives | The political and economic interest of COVID-10 vaccines and its development [34]. |
| Conspiracy theory | Claims and discussion of various theories related to the COVID-19 vaccine and its malicious goals |
| Mandatory vaccine and ethics | Claims related to concerns about mandatory vaccine and ethics. |
| Vaccine reagents | Claims pertaining to concerns about what materials have been used in vaccine |
| Morbidity and mortality | Posts containing claims about morbidity and mortality due to participation in vaccine trials or receiving vaccine in future |
| Vaccine alternatives | Content related to the alternative options to vaccine |
| Susceptibility due to other vaccine exposure | State or claims related to exposures to other vaccine and vulnerability to COVID-19 |
| Miscellaneous | Statements, claims and discussion that did not fit to the above sub-themes |

### Ethics

Only publicly available information was collected. The study protocol was reviewed and approved by the institutional review board of the University of New South Wales ethical review committee (HC200739), Sydney, Australia.

## Results

The surveillance identified 637 rumors and conspiracy theories related to COVID-19 vaccine in 24 languages from 52 countries (Fig 1). Of the total, 91% (578/637) were classified as rumors and 9% (59/637) as conspiracy theories. These items included news articles, social media narratives, online reports and/or blogs that approximately 103.3 million people had liked, shared, reacted to with an emoji, or retweeted on social media. Of all the items, 15% (94/637) were reported in the United States, 13% (82/637) in India, and 12% (77/637) in Brazil. The African continent reported the least (Fig 2). Facebook was the most prevalent (61%) media, followed by Twitter (28.6%) (Fig 3).

The surveillance identified three waves of COVID-19 vaccine rumors and conspiracy theories between 10 January to 30 November 2020. The first wave was between February and May 2020, the second wave between June and September 2020, and the third wave between 16 October and 20 November 2020 (Fig 4). The second wave had the most rumors and conspiracy theories when compared with the first and the third wave. During the initial months of COVID-19, most of the COVID-19 vaccine claims were related to pre-pandemic vaccine and conspiracy theories. More recent claims were related to efficacy and effectiveness of the vaccine, morbidity, and mortality due to participation in the vaccine trial. The three waves of rumors and conspiracy theories we detected were consistent with the three waves of Google Search on COVID-19 vaccine (Fig 3).

### Rumors

Of the 578 items classified as rumors, we further classified them according to sub-themes Table 2.

### Vaccine trials, development, and access

Of the rumors identified, 39% (227/578) were related to COVID-19 vaccine trials, development, delivery, and access. Of these, 76% (173/227) were identified as false/misinformation.

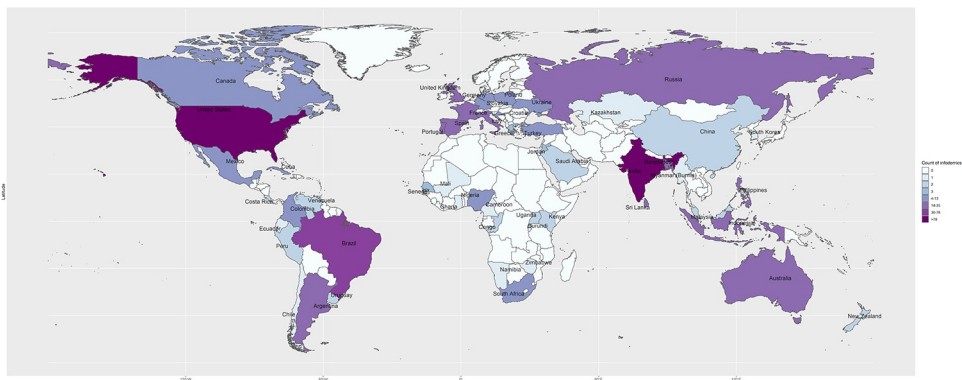

**Fig 1. Global distribution of rumors and conspiracy theories related to COVID-19 vaccine, 31 December 2019–30 November 2020.**

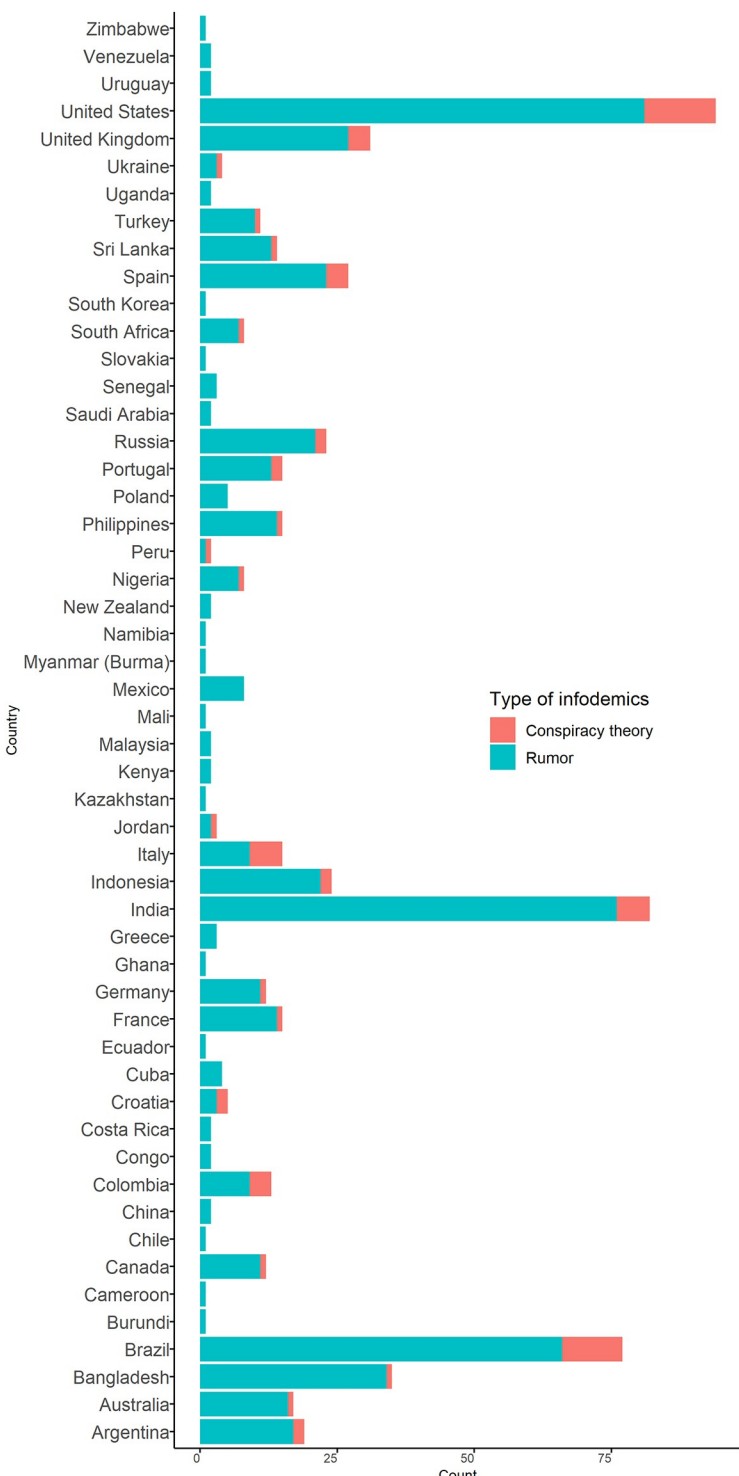

**Fig 2. Rumor and conspiracy theories related to COVID-19 vaccine by country, 31 December 2019–30 November 2020.**

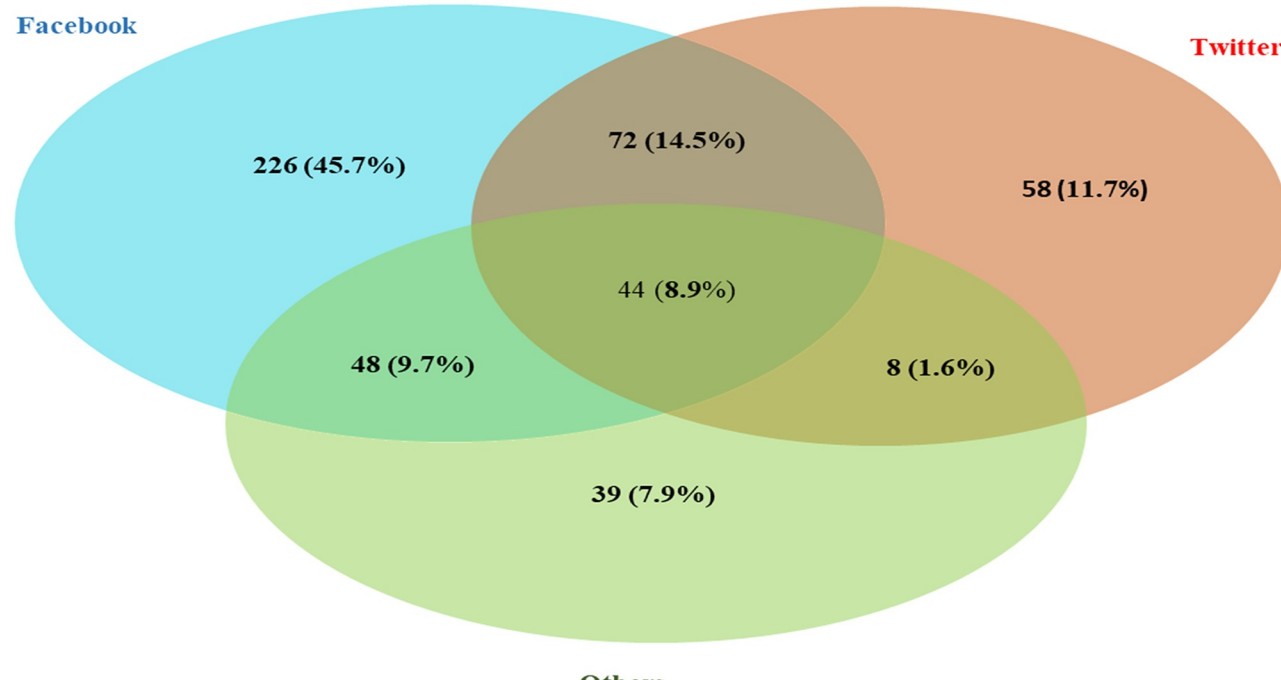

**Fig 3. Rumors and conspiracy theories related to COVID-19 vaccine circulating on different social media platforms, 31 December 2019–30 November 2020.**

Most of these claims described either the start or end of a vaccine trial, the timeline for the roll out of mass vaccination, countries and companies involved in the development process, photos of high-profile people participating in a vaccine trial, and people's access to the vaccine. One of the claims was that Chinese laboratories lacked monkeys, and therefore, the Chinese developed a vaccine that was trialed in Indonesia. Another Facebook post warned people not to be part of the vaccine trial in India. A rumor circulated in Bangladesh that China wanted to use Bangladesh citizens as guinea pigs for a vaccine trial. Another dominant rumor was that crucial phases of the clinical trials were skipped as the pharmaceutical companies would not compensate participants for adverse side effects during the trial. Another commonly circulating rumor was that a Russian vaccine company omitted Phase 3 clinical trials for a COVID-19 vaccine. This claim provoked a lot of concern and criticism from the scientific community that the vaccine was not tested for effectiveness or safety in many humans which might lead to global concern and vaccine hesitancy.

## Morbidity and mortality related to vaccine

The second most common subtheme tracked 22% (125/578) was related to morbidity and mortality due to receiving the COVID-19 vaccine. The most common circulating rumor was that the COVID-19 vaccine would be a messenger Ribonucleic acid (mRNA) vaccine that could alter people's Deoxyribonucleic acid (DNA), subsequently turning them into a genetically modified human being. Other posts postulated that 160 doctors disapproved of the COVID-19 vaccine as it could change human DNA or that it could modify genes, cause cancers, and infertility. Other claims linked the vaccine to crippled children in Africa and to clinical trial participants who had become ill or had severe allergic reactions after being vaccinated.

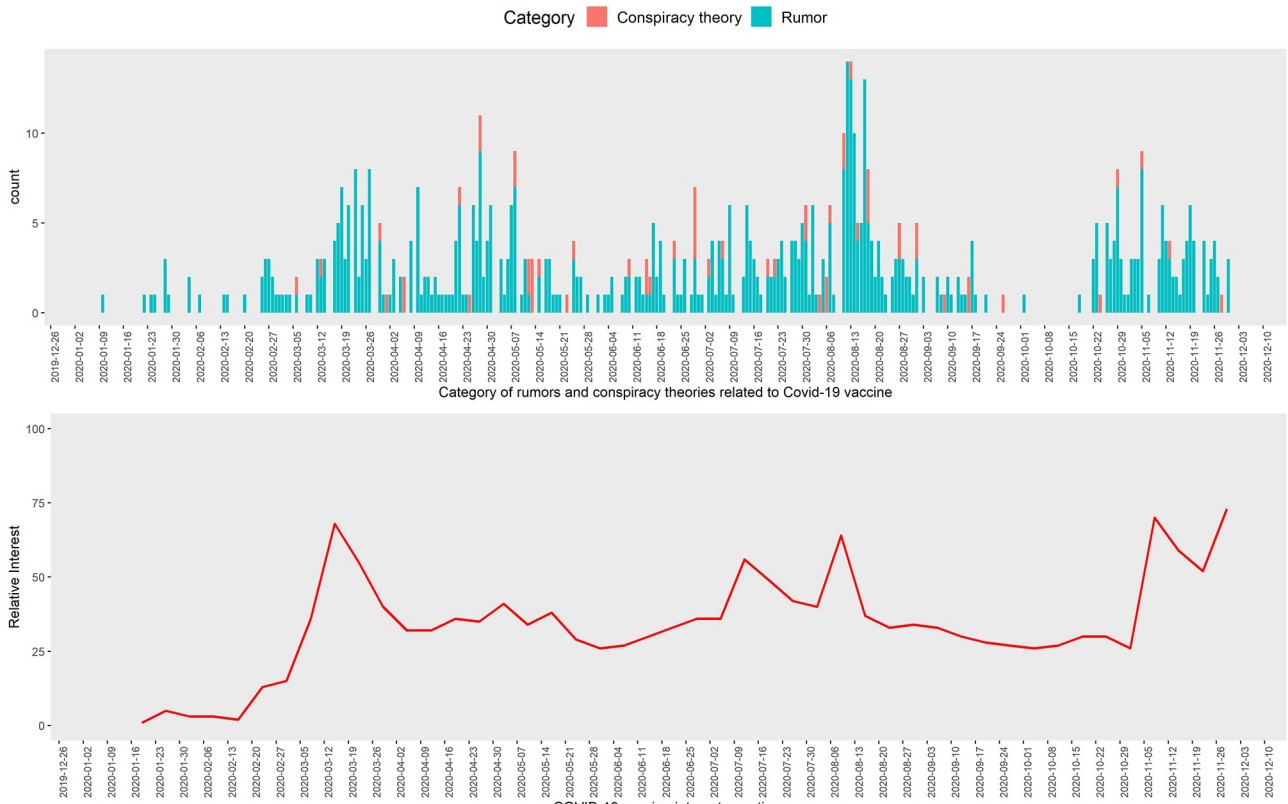

**Fig 4. Spread of misinformation and internet users search interest related to COVID-19 vaccines, 31 December 2019–30 November 2020.** *Google Trends: Numbers represent search interest relative to the highest point on the chart for the given region and time. A value of 100 is the peak popularity for the term. A value of 50 means that the term is half as popular. A score of 0 means there was not enough data for this term.

Some claims stressed that the COVID-19 vaccine was aimed at reducing the world's population, while others raised concerns about vaccine-derived COVID-19 in Russia.

Regarding mortality, some of the claims were related to the estimated deaths due to COVID-19 vaccine and participation in vaccine trials. One of the widely circulated false claims was that the daughter of the Russian president had died after receiving the second dose of COVID-19 vaccine. We also identified other misinformation related to children and soldiers dying after receiving the vaccine in multiple countries; it was reported on Facebook that seven children died after receiving the COVID-19 vaccine in Senegal. Lastly, it was falsely reported that Bill Gates had said that 700,000 people would be injured or die from the COVID-19 vaccine.

### Influenza vaccine increase susceptibility to COVID-19 infection

Of the rumors identified, 4% (23/578) claimed people who were vaccinated for seasonal influenza were at higher risk of COVID-19 infection. Some of the claims argued that largest number of COVID-19 deaths occurred in countries with a higher uptake rate of influenza vaccine among the elderly. It was also claimed that people had been injected with the virus during the administration of influenza vaccine during mass campaigns. Referring to an epidemiological study [23], one claim highlighted that persons with COVID-19 infection who received an influenza vaccine in 2019 were more likely to be hospitalized, and once hospitalized, were more likely to die. Although this finding was not statistically significant after adjusting for

**Table 2. Rumors and conspiracy theories related to COVID-19 vaccine circulating online, 31 December 2019–30 November 2020.**

| Categories | Claims |
|---|---|
| Vaccine development, availability, and access | Indonesia is the only country that has been the guinea pig of the COVID-19. Vaccine from China, even though China does not want to test the vaccine on its people. |
| | COVID-19 vaccine was tested in Indonesia because Chinese laboratories lacked monkeys. |
| | Putin's daughter was among the volunteers for COVID 19 vaccine trials. |
| | Dr Anthony Fauci, the nation's leading infectious disease expert, has made a statement" the COVID-19 vaccine should be rushed through development without proper clinical trials". |
| | Oxford vaccine against COVID-19 will be ready in September. |
| | COVID-19: Vaccine manufacturers are granted indemnification. |
| Safety, efficacy, and acceptance | COVID-19 vaccine will cause infertility. |
| | COVID-19 vaccine may not be effective and has serious side effects. |
| | Bill Gates explains that the COVID vaccine will use experimental technology and permanently alter your DNA. |
| | WHO admits COVID vaccines will not work. |
| | Coronavirus vaccine is ready, able to cure a patient within 3 hours. |
| | The COVID-19 vaccine "contains DNA modifiers and multipliers. |
| | Anthony Fauci remarks that vaccines are toxic and can make you worse. |
| | 160 doctors disapprove of the COVID vaccine because it will change our DNA. |
| | Every fourth American is against taking COVID-19 Vaccine. |
| | One in six people will refuse a coronavirus vaccine. |
| Susceptibility due to influenza vaccine | Flu vaccines make people more vulnerable to infections. |
| | People vaccinated for seasonal influenza face higher chances of catching SARS-COV-2, which causes COVID-19. |
| | Flu vaccines increase the odds of getting COVID-19. |
| | Flu vaccine makes kids more vulnerable to coronaviruses. |
| | Flu Vaccine Kills COVID Patients. |
| Morbidity and mortality | Two children die from COVID-19 Vaccine in Guinea. |
| | WHAT??? French Doctors Kill Two in Deadly Vaccine Trials. |
| | COVID-19 patients died in Guinea after taking a trial vaccine. |
| | Microsoft founder-turned-philanthropist Bill Gates "admits" in an interview that a COVID-19 vaccine could kill or injure up to 700,000 people. |
| | The COVID-19 vaccine will kill 50 million Americans. |
| | American Vaccine against COVID-19 in Ukraine killed 30 per cent of patients. |
| | All participants in a COVID-19 vaccine trials had side effects. |
| | Russian President Vladimir Putin's daughter died on August 15th, 2020, after taking a dose of a new COVID-19 vaccine. |
| | Seven children in Senegal died after being given a vaccine for coronavirus. |
| | My friend just buried 3 children after getting the Vaccine from China. |
| | One of the first volunteers of a COVID-19 vaccine trial in the UK, died. |
| | Vaccines to prevent COVID-19 will modify genes and cause male infertility. |

(*Continued*)

**Table 2.** (Continued)

| Categories | Claims |
|---|---|
| Conspiracy theories | Certificate of Identification of Vaccination with Artificial Intelligence. |
| | With the introduction of the COVID-19 vaccine, microchips (nano-chips) will also be introduced into the human body, then 5G networks will enter the business, through which the world elite will send various signals to the chips, thereby controlling humanity. |
| | The COVID-19 virus is an excuse to inject us with a vaccine that will synchronize us to a digital identity that will be issued in the Civil Registry. |
| | Bill Gates, the founder of harmful vaccination campaigns in developing countries, now plans to use COVID-19 vaccines to surveil the population. |
| | The BBC published a travel piece stating that there will be microchips in the future vaccines funded by the Bill Gates so they can track who has been vaccinated for coronavirus. |
| | COVID-19 is a plan to kill older adults with contaminated vaccines. |
| Political and economic motives | Pharmaceutical companies withheld positive news about its coronavirus vaccine until after the election. |
| | After vaccination against COVID-19, the population of the planet will decrease to 1 billion. |
| | Joe Biden and running mate Kamala Harris are spreading anti-vaccine conspiracy theories" about a COVID-19 vaccine. |
| | Bill Gates vaccine new world order. |
| | Canada will introduce a "World Debt Reset Program," which will set personal debts to zero if a person agrees to get vaccinated for COVID-19 and COVID-21. |
| Vaccine alternatives | Paradigm Shift: US government relies on alternative therapies instead of vaccinations! |
| | COVID-19 will leave the country on its own: health minister. |
| | COVID-19 vaccines are no different from the flu vaccine. |
| | With the survival rate of COVID-19 close to 100% without a vaccine, what exactly will the purpose of the new vaccine be? |
| | One cube sugar with polio vaccine will be effective for corona treatment. |
| | The anti-tuberculosis Bacillus Calmette-Gurin (BCG) vaccine protects people against coronavirus infection. |
| Vaccine reagents | Coronavirus vaccine is made from cells of aborted fetuses. |
| | The vaccine being tested in Brazil is produced with cells from aborted fetuses. |
| | The Covid-19 Vaccine contains toxic ingredients. |
| | Humans will be injected with monkey and pig genes with the Covid-19 Vaccine. |
| Mandatory vaccine and ethics | New Zealand Government can force citizens to be vaccinated. |
| | Vaccinations violate human rights under the principles of the Nuremberg Code. |
| | Doria has banned hydroxychloroquine against covid-19 and wants to force people to get an experimental vaccine. |
| | Biden COVID-19 task force member recommends withholding rent, food stamps from those who refuse the vaccine. |

confounders, some people highlighted these partial results that then became viral on multiple social media platforms. Another false claim was that the risk of infection of COVID-19 increased by 36% due to the influenza vaccine [24].

## Vaccine safety, efficacy, and acceptance

Nine percent (50/578) of rumors were related to safety, efficacy, and acceptance of COVID-19 vaccine. For example, the natural survival rate from COVID-19 without the vaccine would be greater than the effectiveness of the vaccine. Some claims mentioned that COVID-19 vaccine

is a 'total hoax'. A few argued that SARS-COV-2 is a rapidly changing virus, and therefore, the vaccine may not be effective against the future strains of the virus. One claim mentioned that the WHO had admitted the COVID-19 vaccine would not work; another claimed the vaccine would have 80% adverse effects. Regarding acceptance, it was claimed that one in every four Americans was against the vaccine, and that the former US President had urged Africans not to receive vaccines from the USA or UK.

## Vaccine reagents

Six percent (32/578) of rumors were related to COVID-19 vaccine reagents. Of these, 11claimed that the COVID-19 vaccines were being produced with cells from aborted fetuses with one citing the presence of aborted fetus cells in a vaccine producing laboratory in Brazil. Other claimed that the vaccine contained genes from monkeys and pigs. There were also reports that the vaccine contained aluminum that could cause Alzheimer's disease.

## Vaccine alternatives and necessity

Four percent (24/578) of rumors were related to vaccine alternatives and necessity. Many posts claimed that the virus would go away naturally, with no need for a vaccine. One of the posts claimed that the US Government was relying on hydroxychloroquine as an alternative therapy to the vaccine. There were also claims about natural immunity being the best defense against COVID-19, and that eating beef was the best vaccine against COVID-19. Other posts also claimed that the pneumococcal vaccine, Haemophilus influenza type B (Hib) vaccine, the Tuberculosis vaccine (BCG), or one cube of sugar containing the polio vaccine could protect against coronavirus infection.

## Political and economic motives

Two percent (14/578) of rumors were linked to political or economic interest related to the COVID-19 vaccine. The most popular rumor related to economic motives postulated that the vaccine was invented before the COVID-19 pandemic to advance vaccine sales. There were reports that big pharmaceutical companies were negotiating with social media companies, national and international health agencies, newspapers, and television channels to raise vaccine demand among the public for their own financial gain through increased vaccine sales. Other claims were that COVID-19 vaccine development was used as a weapon during the 2020 election campaign in USA and big pharmaceutical companies withheld the positive news about vaccine development until the election results. Another rumor claimed that a free COVID-19 vaccine would be available to those voting for the ruling party in India.

## Conspiracy theories

Of the 59 items classified as conspiracy theories, 97% (57/59) were false. The most popular conspiracy theory circulating via the online platforms was that the COVID-19 vaccine could monitor the human population and take over the world. One theory proposed was the COVID-19 vaccine would contain a microchip through which biometric data could be collected, and large businesses could send signals to the chips using 5G networks, thereby controlling humanity. Another conspiracy theory was that vaccination against COVID-19 was intended to genetically modify humans.

## Discussion

This study identified numerous rumors and conspiracy theories that have the potential to negatively impact the confidence of populations towards the COVID-19 vaccine. In the absence of fact-based information, the circulation of these rumors on multiple social media platforms has the potential to be misinterpreted as credible information. Herd immunity against misinformation and conspiracy theories is required to ensure herd immunity against SARS-CoV-2; the virus causing COVID-19 [5]. It is estimated that the vaccine coverage needs to be approximately 55% to 88% in order to achieve herd immunity against SARS-CoV-2 [25]. However, variations across countries in terms of 'willingness to receive the COVID-19 vaccine' have already been documented [26].

While acknowledging that numerous contextual, social and vaccine related factors can influence vaccine acceptance, including perceptions of the likelihood of getting the disease and risk perceptions towards the vaccine [27], the unsubstantiated claims of morbidity and mortality related to COVID-19 vaccine circulating online may affect COVID-19 vaccine confidence [13, 28]. A national survey conducted among US adults in September 2020 on willingness to get COVID-19 vaccine found a 21% decline when compared with another national survey conducted in May 2020 among similar groups. This decline could be attributable to the exposure to COVID-19 vaccine misinformation on social media [29–32]. In another study, conducted among parents in Australia, 24.3% (458/2018) participants were unsure or not willing to accept a COVID-19 vaccine. Of these, 89% were concerned about vaccine efficacy and safety, and 27% did not believe a COVID-19 vaccine was necessary [33].

For a novel vaccine such as COVID-19 to be successful, the safety and efficacy of the vaccine and its wide acceptance needs to be ensured. As COVID-19 vaccine trials are advancing, so are numerous claims about COVID-19 vaccine safety and efficacy, some of which are false and misleading. While some national and international health agencies and fact-check organizations debunked these claims, the time gap between tracking and debunking misinformation, and its limited reach, may have left some populations vulnerable to vaccine hesitancy [34].

Prior studies showed political beliefs and attitudes towards vaccines are well connected [35, 36]. Our findings that the COVID-19 vaccine has been used for economic and political interest is consistent with studies focused on vaccine rumors conducted prior to 2020 [37]. Rumors that focus on vaccine development delays, or that vaccines will only be freely available to ruling government supporters, may create distrust between government stakeholders and the public. This could ultimately affect any vaccine-related policy implementation. Rumors about the use of vaccination campaigns for political purposes are not new, and such rumors have previously affected vaccination campaigns in some countries, such as Pakistan [38]. During a public health emergency, when there is considerable scientific uncertainty, and political leaders act as frontline crises managers, it often becomes challenging to ensure evidenced-based recommendation over political interest [35]. For example, the statement, "Whether the vaccine comes or not, coronavirus will leave the country" made by a health minister is likely to confuse the public about the necessity of the vaccine [39]. It thus may affect vaccine acceptancy.

The claim that vaccine against COVID-19 had already been invented may lead some to come to the conclusion that this pandemic was being used in an attempt to further vaccine sales [40, 41]. Such misinformation may lead to mistrust on pharmaceutical companies, healthcare agencies and emergency responders and impact overall vaccine acceptancy [42]. Misinformation such as emergence of vaccine-derived SARS-COV-2 virus, or that influenza vaccine can increase the risk of COVID-19 infection, has the potential not only to negatively affect COVID-19 vaccine acceptance, but may also have detrimental impact on confidence in the influenza vaccine, as was identified in Australia [43]. This claim was based on the finding

of a study conducted by Wolf (2020), who noted that people who received the influenza vaccine were 1.36 times more likely to develop respiratory disease caused by a seasonal coronavirus [24].

The rumors that COVID-19 vaccine contains cells from aborted fetus or genes from pigs raised religious concern among the Muslims and Jewish communities. In response to such claims, several ministries in Indonesia were said to slow down the rollout of COVID-19 vaccines unless they received a halal certification to ensure the vaccine is permissible under Islamic law [44]. A similar rumor that the measles and rubella vaccines contained ingredients from pigs, and therefore was not permissible in Islam, resulted in a sharp decline in vaccine acceptancy in Indonesia [45]. The commentary around the use of aborted fetus cells in COVID-19 vaccine may also affect vaccine acceptance among Christians and Buddhists [46]. To combat these rumors, communication materials must be co-designed with the communities of interest, including religious and community leaders, relevant community groups and members. The traditional and religious community leaders can promote immunization by providing practical information, such as vaccination session locations and schedules, during community announcements and after religious services.

The Google Trend analysis showed that there was a high demand for COVID-19 vaccine related information on online platform. The waves of rumors we detected corresponded with the waves of online users' search interest on COVID-19 vaccine information.

There might be a low supply of evidenced-based COVID-19 vaccine information that allowed the opportunity to generate and spread misinformation [34]. Like other infodemics, the COVID-19 vaccine rumor is highly contagious and can spread exponentially around the world [47]. Development of interventions that target individual, community, cultural and societal-level factors are needed to protect people against rumors and conspiracy theories that flatten the misinformation curve. Online platforms and broadcasting channels, including radio, television, and cable channels, should be targeted to promote risk communication and community engagement. According to Tim Nguyen, the head of high impact events preparedness unit at WHO, "*You need to have a certain degree of good information out there to reach populations so that they are inoculated and not susceptible to fake news or disinformation. . .we believe we need to vaccinate 30% of the population with good information in order to have a certain degree of herd immunity against misinformation*" [48]. Websites that provide evidence-based information on COVID-19 vaccines should be developed and disseminated globally as the use of such trusted sources of information have been found effective against misinformation and conspiracy theories [11].

Rumors may also spread offline, for example, word of mouth, and social listening could be utilized to track offline messages. It has been recommended that people should fact check with their family and friends [48]. The WHO recommends a whole-of-society approach to reach diverse communities. Following the bottom-up approach, youth, religious leaders, community stakeholders, faith-based organizations and schools can be engaged to co-design culturally compelling and context-appropriate risk communication and community engagement strategies [48]. International and national health organizations, fact-checking agencies, and infodemic managers can help to identify correct information, and the community ambassadors can diffuse and amplify this information in the community [48].

Moreover, public health agencies should monitor and track the most frequently shared COVID-19 vaccine misinformation on social media [12]. Risk communication targeting disease risks, the role of a vaccine in reducing COVID-19 related morbidity and mortality and known side effects of a vaccine can be posted on social media to improve health literacy among the public.

This study has several limitations. We primarily relied on fact-check and health agency websites, national and international news agencies, and TV channels. All the countries affected by COVID-19 misinformation may not have had all these platforms active during our data collection period. Also, our study may not have detected all misinformation and conspiracy theories circulating online in local native languages. Therefore, the prevalent rumors and conspiracy theories detected through our surveillance could have underestimated the true prevalence. However, we believe our surveillance was timely and covered a reasonable period to enlist vaccine related rumors and conspiracy theories. Secondly, we were unable to track the network of rumors circulating online. For example, many of the rumors we detected were shared multiple times and we did not track further spread of these posts on online platforms. Thirdly, in retrospect, most of the misinformation and conspiracy theories have been debunked by fact-check agencies and national and international health agencies over time. Thus, we do not know if the corrected information changed people's original perceptions. Finally, as COVID-19 science has been advancing, so information related to vaccine has also been changing, and therefore, some items might be misclassified as facts and vice versa.

In conclusion, our study provided a snapshot of rumor and conspiracy theory patterns circulating online that have the potential to build public distrust in the vaccines. The diverse amount of circulating COVID-19 vaccine misinformation could undermine the universal rollout of the COVID-19 vaccine candidates. We suggested that traditional methods of risk communication and community engagement should be explored to track and fact-check misinformation as ways to immunize people against misinformation and thereby pre-empt potential vaccine program disruptions [49]. Policymakers should consider these findings to devise risk communication and community engagement strategies to address these concerns with evidenced-based information. Additionally, topic modelling, artificial intelligence and machine learning technologies have the potential to track and analyze large media data in real-time; however, these technologies could be expensive for low-and middle-income countries.

## Supporting information

**S1 Data.**
(CSV)

## Acknowledgments

We are grateful to the governments of Bangladesh, Canada, Sweden, and the United Kingdom for providing core/unrestricted support to the International Centre for Diarrhoeal Disease Research, Bangladesh (icddr,b), the home institution of the primary author. We are also grateful to the fact-check agencies and other organizations for making available of the COVID-19 vaccine misinformation and conspiracy theories related data online. We would also like to thank Google for using their public domain website-Google Trends(https://trends.google.com/trends/).

## Author Contributions

**Conceptualization:** Md Saiful Islam, Dorothy L. Southern, Sazzad Hossain Khan, Abrar Ahmad Chughtai, Nusrat Homaira, Holly Seale.

**Data curation:** Md Saiful Islam, Abu-Hena Mostofa Kamal, Alamgir Kabir, Sazzad Hossain Khan, S. M. Murshid Hasan, Tonmoy Sarkar, Shayla Sharmin, Shiuli Das, Tuhin Roy, Md Golam Dostogir Harun, Abrar Ahmad Chughtai.

**Formal analysis:** Md Saiful Islam, Abu-Hena Mostofa Kamal, Alamgir Kabir, Dorothy L. Southern, S. M. Murshid Hasan, Tonmoy Sarkar, Shayla Sharmin, Shiuli Das, Abrar Ahmad Chughtai, Nusrat Homaira, Holly Seale.

**Investigation:** Abu-Hena Mostofa Kamal.

**Methodology:** Md Saiful Islam, Alamgir Kabir, Nusrat Homaira, Holly Seale.

**Project administration:** Abu-Hena Mostofa Kamal, Sazzad Hossain Khan, S. M. Murshid Hasan, Tonmoy Sarkar, Shayla Sharmin, Shiuli Das, Tuhin Roy, Md Golam Dostogir Harun, Nusrat Homaira.

**Supervision:** Holly Seale.

**Validation:** Alamgir Kabir, Dorothy L. Southern, Abrar Ahmad Chughtai, Nusrat Homaira, Holly Seale.

**Writing – original draft:** Md Saiful Islam.

**Writing – review & editing:** Md Saiful Islam, Abu-Hena Mostofa Kamal, Alamgir Kabir, Dorothy L. Southern, Sazzad Hossain Khan, S. M. Murshid Hasan, Tonmoy Sarkar, Shayla Sharmin, Shiuli Das, Tuhin Roy, Md Golam Dostogir Harun, Abrar Ahmad Chughtai, Nusrat Homaira, Holly Seale.

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
