## [Decision Letter · Decision Letter 0]

15 Apr 2021

PONE-D-21-06301

COVID-19 vaccine rumors and conspiracy theories: The need for cognitive inoculation against misinformation to improve vaccine adherence

PLOS ONE

Thank you for submitting your manuscript to PLOS ONE. After careful consideration, we feel that it has merit but does not fully meet PLOS ONE’s publication criteria as it currently stands. Therefore, we invite you to submit a revised version of the manuscript that addresses the points raised during the review process.

We look forward to receiving your revised manuscript.

Kind regards,

Luigi Lavorgna

Academic Editor

PLOS ONE

Journal Requirements:

Thank you for stating the following in the Competing Interests section:

The authors have declared that no competing interests exist.

We note that one or more of the authors are employed by a commercial company: Independent Researcher, Sydney, Australia

2a,  Please provide an amended Funding Statement declaring this commercial affiliation, as well as a statement regarding the Role of Funders in your study. If the funding organization did not play a role in the study design, data collection and analysis, decision to publish, or preparation of the manuscript and only provided financial support in the form of authors' salaries and/or research materials, please review your statements relating to the author contributions, and ensure you have specifically and accurately indicated the role(s) that these authors had in your study. You can update author roles in the Author Contributions section of the online submission form.

2b. Please also provide an updated Competing Interests Statement declaring this commercial affiliation along with any other relevant declarations relating to employment, consultancy, patents, products in development, or marketed products, etc. 

We note that Figure 1 in your submission contain map images which may be copyrighted. All PLOS content is published under the Creative Commons Attribution License (CC BY 4.0), which means that the manuscript, images, and Supporting Information files will be freely available online, and any third party is permitted to access, download, copy, distribute, and use these materials in any way, even commercially, with proper attribution. For these reasons, we cannot publish previously copyrighted maps or satellite images created using proprietary data, such as Google software (Google Maps, Street View, and Earth). For more information, see our copyright guidelines: http://journals.plos.org/plosone/s/licenses-and-copyright.

3a, You may seek permission from the original copyright holder of Figure 1 to publish the content specifically under the CC BY 4.0 license. 

3b, If you are unable to obtain permission from the original copyright holder to publish these figures under the CC BY 4.0 license or if the copyright holder’s requirements are incompatible with the CC BY 4.0 license, please either i) remove the figure or ii) supply a replacement figure that complies with the CC BY 4.0 license. Please check copyright information on all replacement figures and update the figure caption with source information. If applicable, please specify in the figure caption text when a figure is similar but not identical to the original image and is therefore for illustrative purposes only.

Reviewers' comments:

Reviewer's Responses to Questions

**Comments to the Author**

1. Is the manuscript technically sound, and do the data support the conclusions?

Reviewer #1: Yes

Reviewer #2: Yes

2. Has the statistical analysis been performed appropriately and rigorously? 

Reviewer #1: Yes

Reviewer #2: Yes

3. Have the authors made all data underlying the findings in their manuscript fully available?

Reviewer #1: Yes

Reviewer #2: Yes

4. Is the manuscript presented in an intelligible fashion and written in standard English?

Reviewer #1: Yes

Reviewer #2: Yes

5. Review Comments to the Author

Reviewer #1: interesting and large integrated report from social media scenario regarding COVID vaccination. Risk of misinformation may lead to important delay in institutional strategy to limit COVID pandemic. I suggest to implement in the discussion the impact of social media influence on population suffering from chronicities focused on coping and fake news and how patients have increasingly been searching for health information on the Internet (see Lavorgna L et al Interact J Med Res. 2017, Lavorgna L et al Mult Scler Relat Disord 2018 ). Figure 3 could be corrected (twiter), but I suggest switching to a better way to show the data (numbers alone poor appealing)

Reviewer #2: The manuscript by Md Saiful Islam et al examine COVID-19 vaccine rumors circulating on online platforms and review interventions to manage misinformation and increase vaccine acceptance. There is emerging interest on the role of internet and social media in influencing the effectiveness of programs, campaigns and initiatives aimed at citizens' health, awareness and well-being; therefore, the article is highly topical. It is well-structured and well written; methods are sound. I only have some minor suggestions to the authors.

- Nowadays, social media have become part of people's daily activities. They facilitate the exchange of information and allow people to share interests, information and personal experiences, overcoming daily limitations of space and time. Therefore, I suggest introducing the role of internet and social media as source of information for patients to highlight the relevance of the paper. (Suggested references: Moccia M, Brigo F, Tedeschi G, Bonavita S, Lavorgna L. Neurology and the Internet: a review. Neurol Sci. 2018 Jun;39(6):981-987. Doi: 10.1007/s10072-018-3339-9. Epub 2018 Mar 28. PMID: 29594831).

- I would also suggest adding in discussion further examples concerning possible solutions to the spread of misinformation through social media. (Suggested references: Lavorgna L, De Stefano M, Sparaco M, Moccia M, Abbadessa G, Montella P, Buonanno D, Esposito S, Clerico M, Cenci C, Trojsi F, Lanzillo R, Rosa L, Morra VB, Ippolito D, Maniscalco G, Bisecco A, Tedeschi G, Bonavita S. Fake news, influencers and health-related professional participation on the Web: A pilot study on a social-network of people with Multiple Sclerosis. Mult Scler Relat Disord. 2018; Waszak PM, Kasprzycka-Waszak W, Kubanek A. The spread of medical fake news in social media – The pilot quantitative study. Health Policy and Technology. 2018 Jun;7(2):115–118. doi: 10.1016/j.hlpt.2018.03.002.)

- Page 19: “COVDI-19”: amend

6. PLOS authors have the option to publish the peer review history of their article (what does this mean?). If published, this will include your full peer review and any attached files.

Reviewer #1: No

Reviewer #2: No

---

## [Author Response · Author response to Decision Letter 0]

27 Apr 2021

PONE-D-21-06301

26-Apr-2021

To

Luigi Lavorgna

Academic Editor

PLOS ONE

We are thankful to the editor and the reviewers for their thorough reviews of the manuscript and for allowing us to respond to the comments. Based on the feedback, we revised the manuscript. The manuscript has also been reviewed by a professional English writing consultant for clarity and flow. The following is an itemized list of our specific responses to the editor and each reviewer's comments. We have also highlighted where the changes have been made in the revision. Now, we believe the manuscript is updated, more precise, clear, and informative.

Comments: Journal Requirements

 Response: Thank you. Based on your suggestion, we reviewed the PLOS ONE's style requirements and revised the authors' affiliation and body of the manuscript. 

Comments: Thank you for stating the following in the Competing Interests section:

The authors have declared that no competing interests exist.

We note that one or more of the authors are employed by a commercial company: Independent Researcher, Sydney, Australia

Response: The commercial affiliation did not play a role in our study, and this does not alter our adherence to PLOS ONE policies of sharing data and materials. Now, we have added this statement in the cover letter.

Comments: 2a, Please provide an amended Funding Statement declaring this commercial affiliation, as well as a statement regarding the Role of Funders in your study. If the funding organization did not play a role in the study design, data collection and analysis, decision to publish, or preparation of the manuscript and only provided financial support in the form of authors' salaries and/or research materials, please review your statements relating to the author contributions, and ensure you have specifically and accurately indicated the role(s) that these authors had in your study. You can update author roles in the Author Contributions section of the online submission form.

"The funder provided support in the form of salaries for authors [insert relevant initials], but did not have any additional role in the study design, data collection and analysis, decision to publish, or preparation of the manuscript. The specific roles of these authors are articulated in the 'author contributions' section."

2b. Please also provide an updated Competing Interests Statement declaring this commercial affiliation along with any other relevant declarations relating to employment, consultancy, patents, products in development, or marketed products, etc. 

Within your Competing Interests Statement, please confirm that this commercial affiliation does not alter your adherence to all PLOS ONE policies on sharing data and materials by including the following statement: "This does not alter our adherence to PLOS ONE policies on sharing data and materials." (as detailed online in our guide for authors http://journals.plos.org/plosone/s/competing-interests) . If this adherence statement is not accurate and there are restrictions on sharing of data and/or materials, please state these. Please note that we cannot proceed with consideration of your article until this information has been declared.

Response: As suggested, we updated the Competing Interest Statements and funding statement in the cover letter as “Associate Professor Holly Seale has previously received funding from drug companies for investigator-driven research and consulting fees to present at conferences/workshops and develop resources (Seqirus, GSK and Sanofi Pasteur). She has also participated in an advisory board meeting for Sanofi Pasteur.

This work has been a voluntary contribution from all authors. The commercial affiliation of the authors did not play a role in our study, and this does not alter our adherence to PLOS ONE policies of sharing data and materials.

The funder provided support in the form of salaries for authors [MSI, AMK, AK, DLS, SHK, SMMH, TS, SS, SD, TR, MGDH, AAC, NH, HS], but did not have any additional role in the study design, data collection, and analysis, decision to publish, or preparation of the manuscript. The specific roles of these authors are articulated in the ‘author contributions’ section”.

 Comment:

1. We note that Figure 1 in your submission contain map images which may be copyrighted. All PLOS content is published under the Creative Commons Attribution License (CC BY 4.0), which means that the manuscript, images, and Supporting Information files will be freely available online, and any third party is permitted to access, download, copy, distribute, and use these materials in any way, even commercially, with proper attribution. For these reasons, we cannot publish previously copyrighted maps or satellite images created using proprietary data, such as Google software (Google Maps, Street View, and Earth). For more information, see our copyright guidelines: http://journals.plos.org/plosone/s/licenses-and-copyright.

3a, You may seek permission from the original copyright holder of Figure 1 to publish the content specifically under the CC BY 4.0 license. 

"I request permission for the open-access journal PLOS ONE to publish XXX under the Creative Commons Attribution License (CCAL) CC BY 4.0 (http://creativecommons.org/licenses/by/4.0/). Please be aware that this license allows unrestricted use and distribution, even commercially, by third parties. Please reply and provide explicit written permission to publish XXX under a CC BY license and complete the attached form."

In the figure caption of the copyrighted figure, please include the following text: "Reprinted from [ref] under a CC BY license, with permission from [name of publisher], original copyright [original copyright year]."

3b, If you are unable to obtain permission from the original copyright holder to publish these figures under the CC BY 4.0 license or if the copyright holder's requirements are incompatible with the CC BY 4.0 license, please either i) remove the figure or ii) supply a replacement figure that complies with the CC BY 4.0 license. Please check copyright information on all replacement figures and update the figure caption with source information. If applicable, please specify in the figure caption text when a figure is similar but not identical to the original image and is therefore for illustrative purposes only.

Response: Figure 1 is original; it has not been copied from any published literature. Based on our data, we created figure 1 using R-statistical software. Therefore, we do not require any copyright permission.

Comments: 4. Please review your reference list to ensure that it is complete and correct. If you have cited papers that have been retracted, please include the rationale for doing so in the manuscript text, or remove these references and replace them with relevant current references. Any changes to the reference list should be mentioned in the rebuttal letter that accompanies your revised manuscript. If you need to cite a retracted article, indicate the retracted status in the References list and include a citation and full reference for the retraction notice.

Response: Thank you. We have double-checked the references and did not find any reference that has been retracted.

Reviewers' comments:

Reviewer's Responses to Questions

Comments to the Author

1. Is the manuscript technically sound, and do the data support the conclusions?

Reviewer #1: Yes

Reviewer #2: Yes

2. Has the statistical analysis been performed appropriately and rigorously?

Reviewer #1: Yes

Reviewer #2: Yes

3. Have the authors made all data underlying the findings in their manuscript fully available?

Reviewer #1: Yes

Reviewer #2: Yes

4. Is the manuscript presented in an intelligible fashion and written in standard English?

Reviewer #1: Yes

Reviewer #2: Yes

5. Review Comments to the Author

Reviewer #1:

Comments: Interesting and large integrated report from social media scenario regarding COVID vaccination. Risk of misinformation may lead to important delay in institutional strategy to limit COVID pandemic. I suggest to implement in the discussion the impact of social media influence on population suffering from chronicities focused on coping and fake news and how patients have increasingly been searching for health information on the Internet (see Lavorgna L et al Interact J Med Res. 2017, Lavorgna L et al Mult Scler Relat Disord 2018 ). 

Response: Thank you for the compliment. We have reviewed the articles recommended and updated the introduction and the discussion section. On page 5 in the Introduction, we have added the following lines, "The social media platform has become a common source for health information. During a pandemic, people may use social media to improve their knowledge about the disease, transmission, and prevention mechanisms[1, 2]. Health information circulating on online platforms are often amplified by misinformation, conspiracy theories, and myths that are not always based on scientific evidence [3]. Health information-seeking behavior on online platforms puts users at risk of being exposed to misinformation and conspiracy theories that could potentially threaten public health[4]. See the manuscript with track changes.

Comments: Figure 3 could be corrected (twiter), but I suggest switching to a better way to show the data (numbers alone poor appealing)

Response: Thank you for noticing this typo. We have corrected it. We think, to present the interaction between groups, Venn diagram is a useful tool. We now incorporated the percentages along with the numbers.

Reviewer #2: 

Comments: The manuscript by Md Saiful Islam et al examine COVID-19 vaccine rumors circulating on online platforms and review interventions to manage misinformation and increase vaccine acceptance. There is emerging interest on the role of internet and social media in influencing the effectiveness of programs, campaigns and initiatives aimed at citizens' health, awareness and well-being; therefore, the article is highly topical. It is well-structured and well written; methods are sound. I only have some minor suggestions to the authors.

Response: Thank you for the appreciation.

Comments: Nowadays, social media have become part of people's daily activities. They facilitate the exchange of information and allow people to share interests, information and personal experiences, overcoming daily limitations of space and time. Therefore, I suggest introducing the role of internet and social media as source of information for patients to highlight the relevance of the paper. (Suggested references: Moccia M, Brigo F, Tedeschi G, Bonavita S, Lavorgna L. Neurology and the Internet: a review. Neurol Sci. 2018 Jun;39(6):981-987. Doi: 10.1007/s10072-018-3339-9. Epub 2018 Mar 28. PMID: 29594831).

Response: Thank you. Based on your recommendations, we have now added, ""The social media platform has become a common source for health information. During a pandemic, people may use social media to improve their knowledge about the disease, transmission, and prevention mechanisms[1, 2]. Health information circulating on online platforms are often amplified by misinformation, conspiracy theories, and myths that are not always based on scientific evidence [3, 4]. Health information-seeking behaviour online puts users at risk of accessing misinformation and conspiracy theories that could potentially threaten public health[4]." on page 5.

Comments: I would also suggest adding in discussion further examples concerning possible solutions to the spread of misinformation through social media. (Suggested references: Lavorgna L, De Stefano M, Sparaco M, Moccia M, Abbadessa G, Montella P, Buonanno D, Esposito S, Clerico M, Cenci C, Trojsi F, Lanzillo R, Rosa L, Morra VB, Ippolito D, Maniscalco G, Bisecco A, Tedeschi G, Bonavita S. Fake news, influencers and health-related professional participation on the Web: A pilot study on a social-network of people with Multiple Sclerosis. Mult Scler Relat Disord. 2018; Waszak PM, Kasprzycka-Waszak W, Kubanek A. The spread of medical fake news in social media – The pilot quantitative study. Health Policy and Technology. 2018 Jun;7(2):115–118. doi: 10.1016/j.hlpt.2018.03.002.)

Response: On page 23, we added, "Websites that provide evidence-based information on COVID-19 vaccines should be developed and disseminated globally as the use of such trusted sources of information were found effective against misinformation and conspiracy theories [3].

On pages 23-24, we added," Moreover, public health agencies should monitor and track the most frequently shared COVID-19 vaccine misinformation on social media [4]. Risk communication targeting disease risks, the role of a vaccine in reducing COVID-19 related morbidity and mortality, and known side effects of a vaccine can be posted on social media to improve health literacy among the public.

Comments: Page 19: "COVDI-19": amend

Response: Thank you for notifying this typo. We have now corrected this to “COVID-19”on page 24.

1. Lavorgna L, Ippolito D, Esposito S, Tedeschi G, Bonavita S. A disease in the age of the web: How to help people with Multiple Sclerosis in social media interaction. Mult Scler Relat Disord. 2017;17:238-9. Epub 2017/10/23. doi: 10.1016/j.msard.2017.08.017. PubMed PMID: 29055466.

2. Islam MS, Sarkar T, Khan SH, Mostofa Kamal AH, Hasan SMM, Kabir A, et al. COVID-19-Related Infodemic and Its Impact on Public Health: A Global Social Media Analysis. Am J Trop Med Hyg. 2020. Epub 2020/08/14. doi: 10.4269/ajtmh.20-0812. PubMed PMID: 32783794.

3. Lavorgna L, De Stefano M, Sparaco M, Moccia M, Abbadessa G, Montella P, et al. Fake news, influencers and health-related professional participation on the Web: A pilot study on a social-network of people with Multiple Sclerosis. Mult Scler Relat Disord. 2018;25:175-8. Epub 2018/08/11. doi: 10.1016/j.msard.2018.07.046. PubMed PMID: 30096683.

4. Waszak PM, Kasprzycka-Waszak W, Kubanek A. The spread of medical fake news in social media – The pilot quantitative study. Health Policy and Technology. 2018;7(2):115-8. doi: https://doi.org/10.1016/j.hlpt.2018.03.002.

---

## [Editor Report · Decision Letter 1]

29 Apr 2021

COVID-19 vaccine rumors and conspiracy theories: The need for cognitive inoculation against misinformation to improve vaccine adherence

PONE-D-21-06301R1

We’re pleased to inform you that your manuscript has been judged scientifically suitable for publication and will be formally accepted for publication once it meets all outstanding technical requirements.

Kind regards,

Luigi Lavorgna

Academic Editor

PLOS ONE

---

## [Editor Report · Acceptance letter]

5 May 2021

PONE-D-21-06301R1 

COVID-19 vaccine rumors and conspiracy theories: The need for cognitive inoculation against misinformation to improve vaccine adherence. 

Dear Dr. Islam:

I'm pleased to inform you that your manuscript has been deemed suitable for publication in PLOS ONE. Congratulations! Your manuscript is now with our production department. 

Kind regards, 

on behalf of

Dr. Luigi Lavorgna 

Academic Editor

PLOS ONE